# Effect of Tannic Acid on Antioxidant Function, Immunity, and Intestinal Barrier of Broilers Co-Infected with Coccidia and *Clostridium perfringens*

**DOI:** 10.3390/ani14060955

**Published:** 2024-03-19

**Authors:** Zhengfan Zhang, Pengtao Xu, Chengao Liu, Jing Chen, Bingbing Ren, Encun Du, Shuangshuang Guo, Peng Li, Lanlan Li, Binying Ding

**Affiliations:** 1Engineering Research Center of Feed Protein Resources on Agricultural By-Products, Ministry of Education, Hubei Key Laboratory of Animal Nutrition and Feed Science, Wuhan Polytechnic University, Wuhan 430023, China; zhang850820@hotmail.com (Z.Z.); xupengtao113@126.com (P.X.); lllca0010@163.com (C.L.); 15139433336@163.com (J.C.); qaz123mlp4562022@163.com (B.R.); guo1shuangshuang@163.com (S.G.); pengli@whpu.edu.cn (P.L.); lan_lanli@163.com (L.L.); 2Hubei Key Laboratory of Animal Embryo and Molecular Breeding, Institute of Animal Science and Veterinary Medicine, Hubei Academy of Agricultural Sciences, Wuhan 430064, China; qiaowan77@126.com

**Keywords:** antioxidant, broiler, intestinal barrier, necrotic enteritis, tannic acid

## Abstract

**Simple Summary:**

Necrotic enteritis (NE) is mainly caused by coccidia and *Clostridium perfringens* (CCP), which can induce intestine injury and oxidative stress in broilers. Tannic acids (TA) are natural polyphenolic compounds with anti-bacterial, anti-inflammatory, and anti-oxidation functions. It has been demonstrated that dietary supplementation with hydrolyzable TA has beneficial effects on the growth and antioxidant capacity of broilers. However, the effects of TA on intestinal health and antioxidative function in broilers with NE conditions still need to be clarified. Thus, this study aimed to evaluate the effects of TA on the antioxidant function, immunity, and intestinal barrier in broilers co-infected with CCP. The results showed that the addition of 1000 mg/kg TA to the diet could improve the jejunal barrier, attenuate the inflammatory response of the jejunum, and increase the antioxidant capacity of the liver and jejunum through the activation of the transcription factor Nrf2 downstream of the Nrf2-Keap1 pathway in CCP infected broilers.

**Abstract:**

The purpose of this study was to determine the efficacy of tannic acid on the antioxidative function, immunity, and intestinal barrier of broilers co-infected with coccidia and *Clostridium perfringens* (CCP). A total of 294 1-day-old arbor acres(AA) broilers were divided into three groups: control group (CON), CCP co-infected group (CCP), and 1000 mg/kg TA + CCP co-infected group (CTA). This trial lasted for 28 days. The results showed that the CCP group decreased the activity of glutathione peroxidase (GSH-Px), total superoxide dismutase (T-SOD), catalase (CAT), and total antioxidant capacity (T-AOC) levels and increased the contents of hydrogen peroxide (H_2_O_2_) and malondialdehyde (MDA) in the jejunum (*p* < 0.05). The mRNA levels of GSH-Px3 and CAT in the liver and jejunum, and the mRNA levels of GSH-Px3, SOD, HO-1, and NAD(P)H quinone oxidoreductase I (NQO1) in the liver were down-regulated by CCP challenge (*p* < 0.05). In addition, the Keap1 and Nrf2 mRNA levels in the liver and jejunum, jejunal glutathione S-transferase (GST), and heme-oxygenase-1 (HO-1) were upregulated in the CCP group compared with CON (*p* < 0.05). The mRNA levels of interleukin 8 (IL-8), IL-1β, inducible nitric oxide synthase (iNOS), and interferon γ (IFN-γ) in the jejunum were elevated, and jejunal mRNA levels of IL-10, zonula occludens protein1 (ZO-1), claudin-1, claudin-2, and occludin were decreased in the CCP treatment (*p* < 0.05). Dietary supplementation with 1000 mg/kg TA increased the activity of GSH-Px, T-SOD, CAT, and T-AOC and decreased the contents of H_2_O_2_ and MDA in the jejunum (*p* < 0.05). Compared with the CCP group, TA decreased the mRNA level of Keap1 and Nrf2 in the liver and jejunum, increased the GSH-Px3, SOD, and CAT mRNA in the liver, and alleviated the rise of IL-8, IL-1β, iNOS, and IFN-γ and decrease in IL-10, occludin gene expression in the jejunum (*p* < 0.05). In conclusion, the addition of 1000 mg/kg TA to the diet improved the jejunal barrier, mitigated the jejunal inflammation, and increased the antioxidant capacity of the liver and jejunum through the activation of the transcription factor Nrf2 downstream of the Nrf2-Keap1 pathway in broilers with NE condition.

## 1. Introduction

Necrotic enteritis (NE) is a prevalent infectious gastrointestinal illness in broiler production, predominantly caused by *Clostridium perfringen*. *C. perfringens* is secondary to coccidia (coccidia is primary), or they may be co-infected, resulting in an inflammatory reaction of the intestinal mucosa and oxidative stress, which can cause a tremendous economic loss in the poultry industry [1,2,3,4]. Oxidative stress and inflammation disrupt mucosal cells and tight junction proteins, impeding the self-repair mechanisms of the intestinal barrier [5,6]. Antibiotics protect broilers from intestinal diseases and increase productivity. However, antibiotics in animal feed have been limited or outlawed in many countries due to bacteria and antibiotic resistance. Therefore, it is urgent to find natural and environmentally friendly feed additives to control the growing prevalence of NE in poultry effectively.

Tannins are natural polyphenolic substances in numerous plants with anti-bacterial, anti-viral, anti-inflammatory, and anti-oxidation functions [7,8]. Research has demonstrated that dietary supplementation with hydrolyzed tannins can reduce oxidative stress and improve antioxidant capacities in broilers [9,10]. Dietary TA can also alleviate tissue inflammation and protect intestinal health by lowering the mRNA levels of tissue inflammation elements and increasing the mRNA levels of intestinal intact proteins in broilers [11,12,13]. It has been shown that TA can alleviate the inflammation of broilers with NE conditions and enhance the antioxidative ability by activating the Nrf2-Keap1 signaling pathway in rats [14,15,16,17]. However, whether TA could improve antioxidant capacity and alleviate oxidative stress through the Nrf2-Keap1 signaling pathway in broilers with NE condition has yet to be reported.

Therefore, this study aimed to investigate the efficacy of TA on antioxidant functions, immune, and gut barrier of broilers co-infected with CCP to provide some data references and theoretical backup for protecting TA against poultry with NE condition.

## 2. Materials and Methods

### 2.1. Animal Care and Diet

The study was performed for 28 days in a one-way, fully randomized design. A total of 294 1-day-old AA broilers, with an initial BW of 46.40 ± 0.40 g, were randomized to 3 treatment groups with 7 replicates of 14 broilers in each pen (7 males and 7 females). Treatments were as follows: the control group without CCP challenge (CON), the co-infected group with CCP challenge (CCP), and the 1000 mg/kg TA (≥80% the hydrolysable TA from Chinese gallnut, Wufeng Chicheng Biotechnology Company Limited, Yichang, China) + co-infected group with CCP challenge (CTA). The diets were fed in Phase 1 (Day 1 to Day 14) and Phase 2 (Day 15 to Day 28). The basal diets are formulated concerning and in conjunction with the Arbor Acres Broiler Nutrition Specifications (2019); the compositions and nutrition contents of the basal diets are summarized in Table 1. On days 7 and 10 of the study, broilers in the CCP and CTA groups were orally injected with 1 mL of quadrivalent anti-coccidial vaccine (purchased from Foshan Zhengdian Biotechnology Co., Ltd., Foshan, Guangdong, China). The quadrivalent vaccine consisted of 1 × 10^5^ oocysts of the *E. acervuline* strain PAHY, plus the 5 × 10^4^ oocysts of *E. necatrix* strain PNHZ, *E. tenella* strain PTMZ, and *E. maxima* strain PMHY. The recommended inoculation dose for each bird was 1100 ± 110 sporulated oocysts, and the dose used in the present study was 30 times the recommended dose. Meanwhile, the CON group received an equivalent amount of saline. From day 16 to day 20, birds in the CCP and CTA groups per bird per day were orally given 1 mL of fresh *C. perfringens* (1 × 10^8^ CFU/mL, gavage using a 1 mL pipette gun, the *C. perfringens*, CVCC2030, was obtained from the China Veterinary Microbial Culture Collection and Management Center, Beijing, China), while birds in the CON group were orally administered an equivalent amount of sterile broth. All broilers were kept in cages with free access to feed and water throughout the trial. The ambient temperature was maintained at 33 ± 2 °C for the first week and then progressively lowered to 22 °C until the experiment ended. The broilers were exposed to 23 h of light and 1 h of darkness per day throughout the experimental period.

### 2.2. Sample Collection

On days 14, 21, and 28, two broilers of average weight were chosen in each replicate, which was euthanized by cervical dislocation and then promptly slaughtered for sampling. Liver and jejunum(mid-jejunum) were collected and frozen in liquid nitrogen, then transferred to a freezer at −80 °C for storage.

### 2.3. Antioxidant Indexes

Liver and jejunal samples were placed in a mortar, to which liquid nitrogen was added and then crushed with a pestle. A total of 0.1 g of liver and jejunum samples were placed in 1.5 mL centrifuge tubes, respectively, and 0.9 mL of saline was injected into the sample centrifuge tubes using a pipette, followed by centrifugation in a centrifuge (Fresco 21, Thermo Scientific, Wilmington, DE, USA) for 15 min at 3500× *g* at 4 °C, and the supernatant was collected. Off-the-shelf test kits were purchased from Nanjing Jianjian Bioengineering Institute, Nanjing, China. The activity of GSH-Px, T-SOD, and CAT, the concentration of H_2_O_2_ and MDA, and T-AOC in the liver and jejunum, were tested following the manufacturer’s directions.

### 2.4. RNA Isolation and Quantitative Real-Time PCR

RNA isolation and quantitative real-time PCR of liver and jejunal samples were performed, as reported by Guo and Li [18,19]. The SYBR Premix Ex Taq kit (Takara Biotechnology (Dalian) Co., Ltd., Dalian, China) used cDNA from each sample and primers. The qPCR was performed on an Applied Biosystems 7500 Fast Real-Time PCR System (Foster City, CA, USA). The relative expression of each gene was analyzed using the 2^−ΔΔCt^ method (Relative quantification) [20]. β-actin was used as the endogenous reference gene for all the genes tested. Table 2 lists the sequences of the relevant assay genes and internal reference primers used in this study.

### 2.5. Statistical Analysis

Data from this trial were analyzed by one-way ANOVA conducted using least significant difference (LSD) multiple comparisons using SPSS Statistics 25 (SPSS Inc., Chicago, IL, USA). Data presented as mean ± standard deviation (SD). Statistically significant at *p* < 0.05. The graphs were created using GraphPad Prism 10.0 software (GraphPad Software, LLC, San Diego, CA, USA).

## 3. Results

### 3.1. Antioxidant Indexes

In our previous study [21], we found that the hepatic CAT and T-SOD activity was decreased, and the MDA and H_2_O_2_ contents in the liver were decreased in the CCP group than those in the CON and CTA groups (*p* < 0.05).

Jejunal antioxidant capacity results are shown in Table 3. Jejunal GSH-Px and CAT activities were lower in the CCP group than in the CON and CTA groups on day 14 (*p* < 0.05). The GSH-Px level of jejunum was decreased in the CCP group compared with the CON and CTA, and T-SOD activity in CCP was lower than the CTA on day 21 (*p* < 0.05). Jejunal T-AOC, T-SOD, and CAT levels were lower in the CCP group than in the CON and CTA groups on day 28 (*p* < 0.05). On days 14 and 21, the content of jejunal H_2_O_2_ was increased in the CCP compared with the CON and CTA groups, and MDA content was more significant than the CON, and MDA content on day 21 was markedly decreased in the CTA group (*p* < 0.05).

### 3.2. The mRNA Levels of Antioxidant Enzymes in the Liver

As shown in Figure 1, hepatic GSH-Px1, GSH-Px3, CAT, and SOD mRNA levels on days 14 and 21 were reduced by CCP challenge (Figure 1A,B, *p* < 0.05); GSH-Px3 mRNA levels in the liver on day 28 were markedly greater in the CON group than in the CTA and CCP groups (Figure 1C, *p* < 0.05). The CTA could alleviate the downregulation of hepatic GSH-Px1, CAT, and SOD mRNA levels of broilers challenged by CCP (*p* < 0.05).

### 3.3. The mRNA Levels of Antioxidant Enzymes in the Jejunum

The results of jejunal antioxidant enzyme mRNA levels are shown in Figure 2. Compared to the CON and CTA groups, the CCP significantly lowered jejunal GSH-Px3 and CAT mRNA levels on day 14 (Figure 2A, *p* < 0.05). The SOD mRNA level in the CTA group on day 21 was markedly higher than those in the CON and CCP groups. Additionally, the CCP significantly downregulated the CAT mRNA level in the jejunum on day 21 compared with the CON and CTA (Figure 2B, *p* < 0.05). There was no significant difference in relevant antioxidant enzymes in the jejunum among treatments on day 28 (Figure 2C, *p* > 0.05).

### 3.4. The mRNA Levels of the Antioxidant Pathway in the Liver

As presented in Figure 3, on day 14, the Nrf2 mRNA level was lower in the CON in the liver than in the CTA and CCP groups, and the GST mRNA level was markedly higher in CTA than in CON and CCP (Figure 3A, *p* < 0.05). On day 21, the CCP significantly increased the Keap1 and Nrf2 mRNA levels and significantly reduced the mRNA levels of HO-1 and NQO1 compared to the CON; compared to the CCP, the CTA markedly reduced the mRNA levels of Keap1 and Nrf2 and significantly increased the mRNA levels of GST and NQO1(Figure 3B, *p* < 0.05). On day 28, the CCP significantly enhanced the mRNA levels of Keap1 and Nrf2 compared to the CON; the CTA markedly decreased the mRNA levels of Keap1, Nrf2, and GST compared to the CCP (Figure 3C, *p* < 0.05).

### 3.5. The mRNA Levels of the Antioxidant Pathway in the Jejunum

The results of jejunal antioxidant pathways mRNA levels are shown in Figure 4. On day 14, the mRNA level of Keap1 was markedly more significant in the CCP in the jejunum than in the CON and CTA; the mRNA level of Nrf2 was markedly lower in the CTA in the jejunum than in COT and CCP (Figure 4A, *p* < 0.05). The mRNA levels of Keap1, Nrf2, GST, and HO-1 in the jejunum on day 21 were upregulated by the CCP challenge (Figure 4B, *p* < 0.05); the mRNA levels of Keap1 and GST in the jejunum on day 28 were increased with CCP challenge (Figure 4C, *p* < 0.05); The CTA could alleviate the upregulation of Keap1, Nrf2, GST and HO-1 in the jejunum on days 21 and 28 in the CCP-challenged broilers (Figure 4B,C, *p* < 0.05).

### 3.6. The mRNA Levels of the Cytokines in the Jejunum

As shown in Figure 5, on day 14, the CCP markedly elevated the mRNA levels of IL-8, IL-1β, iNOS, and IFN-γ in the jejunum compared with the CON and CTA groups (Figure 5A, *p* < 0.05). The CTA markedly elevated the mRNA levels of IL-10 in the jejunum on day 21 compared with the CON and CCP groups (Figure 5B, *p* < 0.05). The mRNA levels of IL-8, iNOS, and IFN-γ in the jejunum on days 21 and 28 were elevated by CCP challenge (Figure 5A,B, *p* < 0.05). Alternatively, the mRNA level of IL-1β was elevated, and the mRNA level of IL-10 was reduced in the jejunum on day 28 with CCP challenge (Figure 5C, *p* < 0.05). The CTA could alleviate the increase in IL-8, IL-1β, iNOS, and IFN-γ mRNA levels and the decrease in IL-10 mRNA level in the jejunum in the CCP-challenged broilers (*p* < 0.05).

### 3.7. The mRNA Levels of Barrier Factors in the Jejunum

According to Figure 6. The ZO-1 mRNA level in the jejunum on day 14 was reduced by CCP challenge in comparison with the CON group; the CCP markedly reduced the mRNA levels of Claudin-2 in the jejunum on day 14 compared to the CON and CTA groups (Figure 6A, *p* < 0.05). Jejunal mRNA levels of ZO-1 and Occludin were markedly reduced by CCP challenge on day 21 compared to the CON group (Figure 6B, *p* < 0.05). The mRNA levels of Claudin-1, Claudin-2, and Occludin in the jejunum on day 28 were significantly reduced by CCP challenge (Figure 6C, *p* < 0.05). The CTA could alleviate the decrease in Claudin-2 and Occludin mRNA in the jejunum in the CCP-challenged broilers (*p* < 0.05).

## 4. Discussion

The NE is a prevalent infectious gastrointestinal illness in broiler production, predominantly associated with *C. perfringens* types A and G. *C. perfringens* is secondary to coccidia (coccidia is primary), or they may be co-infected. This study used the NE model constructed by coccidia and *C. perfringens* type A. In our previous study, we found that the infection of coccidia and *C. perfringens* reduced the growth performance and caused intestinal morphological structure damage in broilers [21]. Therefore, the infection model is well-established. We also found significantly decreased levels of high-density lipoprotein (HDL), gamma-glutamyl transferase (GGT), and lactate dehydrogenase (LDH) in serum in broilers under NE conditions compared with the control group. HDL is associated with lipid metabolism, and GGT and LDH are associated with liver function [21]. Therefore, in this study, we investigated the effects of TA on hepatic antioxidant function and jejunal function in broilers with NE conditions.

The antioxidant system consists of several components, including enzymatic species such as SOD, CAT, and GSH-Px, as well as non-enzymatic species such as T-AOC [22,23]. When animals are exposed to pathogens, viruses, and harsh environments, the body produces large amounts of reactive oxygen species (ROS) and MDA, which may cause oxidative stress [24]. It has been shown that broilers with NE condition reduced antioxidant function, induced inflammatory response, and impaired intestinal barrier function [19,25,26]. Consistent with previous findings, we found that Jejunal H_2_O_2_ and MDA contents were increased, and T-AOC, SOD, CAT, and GSH-Px levels were decreased by CCP challenge, which indicated that CCP co-infection caused oxidative stress in the broiler. Dietary supplementation with TA has been shown to enhance the activities of intestinal GSH-Px, SOD, and CAT, and to decrease H_2_O_2_ and MDA contents, thereby increasing intestinal antioxidant capacity and protecting the intestinal tract from damage caused by oxidative stress [9,10,27]. In the present study, we found that dietary supplementation with 1000 mg/kg TA reduced the increase in peroxides caused by CCP challenge and enhanced the activities of jejunal antioxidant enzymes. In our previous study, dietary supplementation with 1000 mg/kg TA could alleviate the decrease in antioxidant enzyme levels and increase H_2_O_2_ and MDA contents in the liver by CCP challenge [21]. These results indicated that dietary supplementation with 1000 mg/kg TA could alleviate the oxidative stress in the liver and jejunum of broilers caused by CCP challenge.

The Nrf2-Keap1 system is a significant regulatory pathway against oxidative stress [28]. The activation of the Nrf2-Keap1 pathway enhances antioxidant defense factors-associated genes, which include the GSH-Px1, GSH-Px3, SOD, CAT, HO-1, NQO1, and GST [29]. GSH-Px1 and GSH-Px3 belong to selenium-containing GSH-Px, which have the function of cleaning up reactive oxygen species, with the difference that GSH-Px1 exists in the cytoplasm and mitochondria, and GSH-Px3 exists in the extracellular [30,31,32,33]. Invasion by pathogens can upregulate the expression of the Nrf2 and keap1 genes, which causes oxidative stress in the organism and promotes the entry of Nrf2 into the nucleus to participate in the synthesis of several antioxidant enzymes in reaction to oxidative stress [19,34,35]. In our study, Keap1 and Nrf2 mRNA levels were elevated in the liver and jejunum, and jejunal GST and HO-1 mRNA levels were increased in the CCP-challenged broilers, suggesting that CCP infection induces oxidative stress in broilers. However, we found that the mRNA levels of GSH-Px1, CAT, and SOD in the liver and jejunal CAT were decreased, and jejunal H_2_O_2_ and MDA content in broilers with the CCP challenge were increased. It showed that CCP-infected broilers responded but could not successfully increase antioxidant enzymes to scavenge free radicals, causing oxidative damage to the liver and jejunum. Studies have shown that tannins can enhance the mRNA levels of antioxidant enzymes and reduce oxidative stress to protect the health of the body [36,37,38]. Consistent with these studies, we found that dietary supplementation with 1000 mg/kg TA could alleviate the up-regulation of Keap1 and Nrf2 mRNA levels and enhance the mRNA levels of GSH-Px1, SOD, CAT in the liver and jejunal CAT in the CCP-challenged broilers, which suggested that dietary supplementation with 1000 mg/kg TA could enhance antioxidant enzyme activities through the activation of the transcription factor Nrf2 downstream of the Nrf2-Keap1 pathway to alleviate oxidative damage in the liver and jejunum caused in the CCP-challenged broilers.

Intestinal immune barrier homeostasis is maintained by releasing inflammation and anti-inflammation factors by bowel-related lymphoid tissues composed of various cells that prevent pathogen invasion [39]. This study aimed to determine the expression of specific immune genes in the intestine, mainly belonging to pro-inflammatory (IL-8, IL-1β, iNOS, IFN-γ, TNF-α) and anti-inflammatory (TGF-β4, IL-10) mediators. Pro- and anti-inflammatory cytokines are essential to immune response homeostasis and inflammation [40]. It has been found that when inflammation occurs in the gut upon invasion by pathogens, there is a significant elevation in the expression of the pro-inflammatory genes, which may or may not coincide with a reduction in the anti-inflammatory gene expression [18,41,42]. This study found that jejunal pro-inflammatory mediators (IL-8, IL-1β, iNOS, IFN-γ) were significantly upregulated, the anti-inflammatory mediator (IL-10) was significantly downregulated in broilers with the CCP challenge, and these results suggested that CCP-infected broilers elicited an intestinal inflammatory response in the jejunum. In our study, dietary supplementation with 1000 mg/kg TA significantly diminished pro-inflammatory mediators and elevated anti-inflammatory mediators with CCP challenge, consistent with previous studies [43,44,45]. These results suggest that dietary supplementation with 1000 mg/kg TA is beneficial in alleviating intestinal inflammatory responses in broilers co-infected with CCP.

Tight junction proteins are essential for maintaining the epithelial barrier, which maintains the diffusion barrier and closes the cell gap. ZO-1, occludin, and claudins are the most critical proteins of the intestinal barrier [42]. It was found that both inflammatory mediators and cytokines result in abnormal expression of tight junction proteins such as ZO-1, claudin-1, and occludin, elevating intestinal mucosal permeability and impairing intestinal barrier function [46,47,48,49,50]. It was shown that CCP infection of broilers resulted in significant downregulation of tight junction proteins, causing intestinal damage in broilers [51,52]. In the present study, the mRNA levels of ZO-1, claudin-1, claudin-2, and occludin in the jejunum were significantly downregulated in CCP-infected broilers, which suggested that the jejunal barrier is impaired in CCP-infected broilers. Tannins significantly upregulate ZO-1 levels to protect against jejunal barrier damage. Research has shown that dietary TA supplementation can increase the mRNA levels of ZO-1, claudin-1, claudin-2, and occludin to mitigate intestinal barrier damage [53,54]. In line with prior research, we found that dietary supplementation with 1000 mg/kg TA raised the mRNA levels of claudin-2 and occludin, which suggested that dietary supplementation with TA is beneficial in ameliorating jejunal barrier damage in CCP-challenged broilers.

In this study, dietary supplementation with 1000 mg/kg TA could improve liver and jejunal function in broilers with NE conditions. However, we also found that TA increases GST mRNA levels, which may cause some adverse effects on the liver in broilers. The reason may be related to TA’s origin, dose, and degree of polymerization of TA, which are closely related to its bioavailability. Studies have shown that highly polymerized tannins, with high molecular weight, are more poorly absorbed in the small intestine [55], and hydrolyzed tannins are oligomers of gallic acid, ellagic acid, and glucose that are partially hydrolyzed by digestive acids in the small intestine for easier absorption [56,57]. However, adding high doses of TA to diets may negatively affect performance, lymphoid organ weights, and ileal digestibility of amino acids in broilers and cause liver damage in mice [58,59]. Therefore, the increase in the GST mRNA level in the liver in this trial may be related to the dose of TA. We will further investigate the optimal dose of TA added to the ration to improve the health of broilers with NE conditions.

## 5. Conclusions

The co-infection of CCP reduced antioxidant capacity, induced intestinal inflammatory response, and impaired the intestinal barrier functions of broilers. Dietary supplementation with 1000 mg/kg TA could protect the intestinal barrier, mitigate the inflammatory response, and enhance antioxidant capacity by the activation of the transcription factor Nrf2 downstream of the Nrf2-Keap1 pathway, thereby protecting the intestinal health of co-infected broilers with CCP.

## Figures and Tables

**Figure 1 animals-14-00955-f001:**
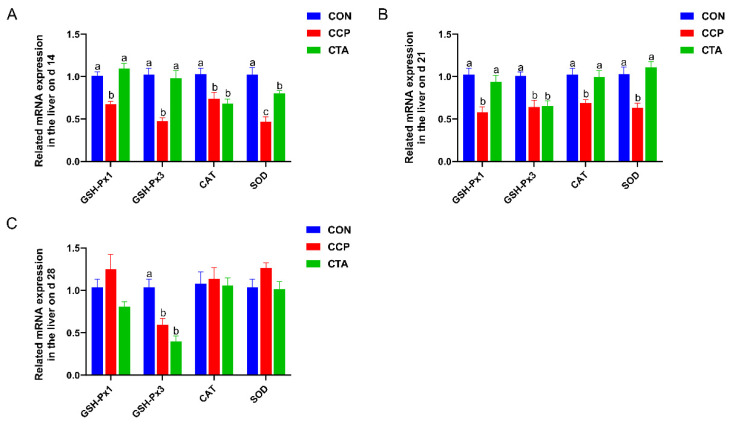
Antioxidant enzymes mRNA expression in the liver ((**A**) on day 14, (**B**) on day 21, (**C**) on day 28). Numbers are expressed as mean and SD, n = 14. ^a,b,c^ Values not sharing a superscript in the same row are markedly different.

**Figure 2 animals-14-00955-f002:**
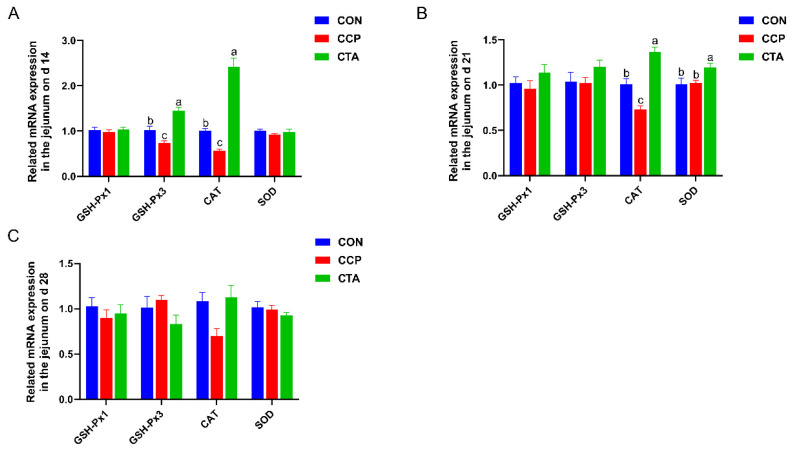
Jejunal antioxidant enzyme mRNA expression ((**A**) on day 14, (**B**) on day 21, (**C**) on day 28). Numbers are expressed as mean and SD, n = 14. ^a,b,c^ Values not sharing a superscript in the same row are markedly different.

**Figure 3 animals-14-00955-f003:**
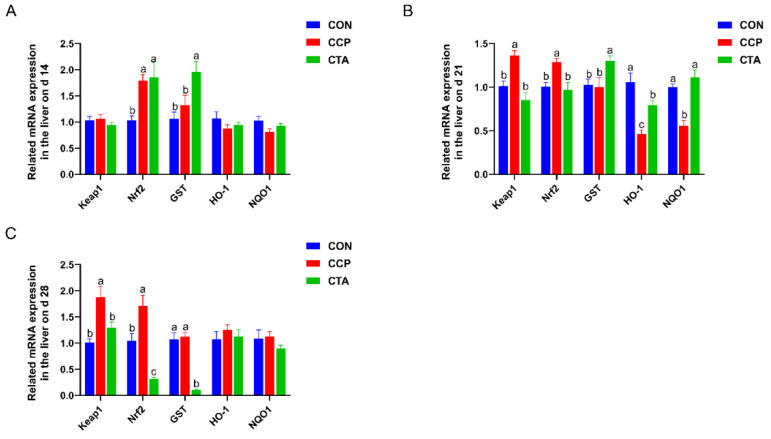
Antioxidant pathway mRNA expression in the liver ((**A**) on day 14, (**B**) on day 21, (**C**) on day 28). Numbers are expressed as mean and SD, n = 14. ^a,b,c^ Values not sharing a superscript in the same row are markedly different.

**Figure 4 animals-14-00955-f004:**
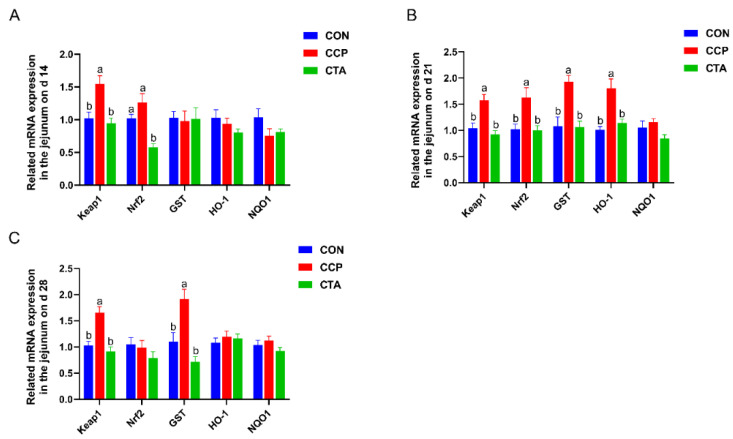
Antioxidant pathway mRNA expression in the jejunum ((**A**) on day 14, (**B**) on day 21, (**C**) on day 28). Numbers are expressed as mean and SD, n = 14. ^a,b^ Values not sharing a superscript in the same row are markedly different.

**Figure 5 animals-14-00955-f005:**
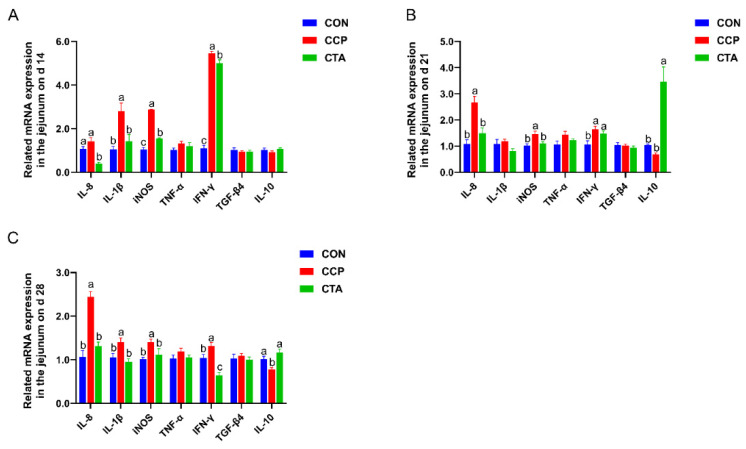
The cytokines mRNA expression in the jejunum ((**A**) on day 14, (**B**) on day 21, (**C**) on day 28). Numbers are expressed as mean and SD, n = 14. ^a,b,c^ Values not sharing a superscript in the same row are markedly different.

**Figure 6 animals-14-00955-f006:**
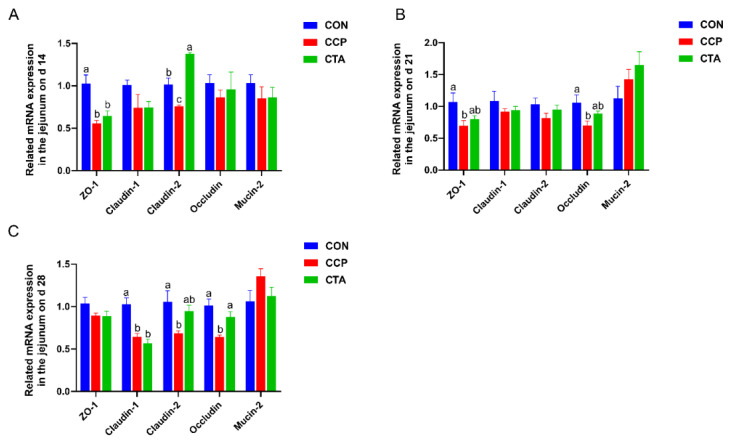
Barrier factors mRNA expression in the jejunum ((**A**) on day 14, (**B**) on day 21, (**C**) on day 28). Numbers are expressed as mean and SD, n = 14. ^a,b,c^ Values not sharing a superscript in the same row are markedly different.

**Table 1 animals-14-00955-t001:** Compositions and nutrient content of the basal diet (%).

Items	1 to 14 Days	15 to 28 Days
Ingredients		
Corn	10.00	10.00
Wheat	55.42	58.90
Soybean meal	23.50	19.37
Fish meal	5.00	5.00
Soybean oil	2.50	3.80
CaHPO_4_	1.05	0.75
Limestone	1.00	0.80
NaCl	0.35	0.35
L-lysine•HCl	0.35	0.25
DL-methionine	0.20	0.15
D-threonine	0.15	0.15
Choline chloride	0.25	0.25
Premix ^a^	0.23	0.23
Total	100.00	100.00
Nutrient levels ^b^		
ME/(MJ/kg)	12.55	12.95
Crude protein	21.53	20.14
Ca	0.96	0.86
Non-phytate phosphorus	0.45	0.40
Digestible lysine	1.25	1.05
Digestible methionine	0.53	0.46
Digestible threonine	0.81	0.76
Digestible tryptophan	0.23	0.22

^a^ The premix provided the following per kg of diets: Cu 10 mg, Zn 100 mg, Fe 80 mg, Mn 100 mg, Se 0.3 mg, I 0.7 mg, VA 12,000 IU, VD_3_ 3000 IU, VK_3_ 3.2 mg, VB_1_ 3 mg, VB_2_ 8.0 mg, VB_12_ 0.025 mg, VE 44 IU, biotin 0.0325 mg, folic acid 2.00 mg, pantothenic acid 15 mg, and nicotinic acid 15 mg. ^b^ Non-phytate phosphorus, ME and digestible amino acids were calculated values, while Ca and CP were measured values.

**Table 2 animals-14-00955-t002:** List of gene primer sequences ^a^.

Gene	Primer Sequence (5′ to 3′)	Product Length	NCBI Number
*β-actin*	F-ACTCTGGTGATGGTGTTAC	497	NM 205518.2
	R-GGCTGTGATCTCCTTCTG		
*Nrf-2*	F-ATCACCTCTTCTGCACCGAA	229	NM 205117.2
	R-GCTTTCTCCCGCTCTTTCTG		
*Keap1*	F-CAACTTCGCCGAGCAGA	179	KU 321503179
	R-CGTGGAACACCTCCGACT		
*GST*	F-AGAGTCGAAGCCTGATGCAC	220	NM_001001777.2
	R-CACTCCGCTTATCAGCAAACA		
*HO-1*	F-ACGAGTTCAAGCTGGTCACG	244	NM_205344.2
	R-GGATGCTTCTTGCCAACGAC		
*NQO1*	F-TCGCCGAGCAGAAGAAGATTGAAG	191	NM_001277621.1
	R-GGTGGTGAGTGACAGCATGGC		
*GSH-px1*	F-GACCAACCCGCAGTACATCA	204	NM_001277853.3
	R-GAGGTGCGGGCTTTCCTTTA		
*GSH-px3*	F-AAGTGCCAGGTGAACGGGAAGG	204	NM 001163232.3
	R-AGGGCTGTAGCGGCGGAAAG		
*SOD*	F-GGTGCTCACTTTAATCCTG	109	NM 205064.2
	R-CTACTTCTGCCACTCCTCC		
*CAT*	F-GGTTCGGTGGGGTTGTCTTT	213	NM_001031215.2
	R-CACCAGTGGTCAAGGCATCT		
*iNOS*	F-CCTGGAGGTCCTGGAAGAGT	82	NM_204961.2
	R-CCTGGGTTTCAGAAGTGGC		
*TNF-α*	F-GAGCGTTGACTTGGCTGTC	64	NM_204267.2
	R-AAGCAACAACCAGCTATGCAC		
*IFN-γ*	F-AGCTGACGGTGGACCTATTATT	259	Y07922.1
	R-GGCTTTGCGCTGGATTC		
*IL-1β*	F-ACTGGGCATCAAGGGCTA	131	NM_204524.2
	R-GGTAGAAGATGAAGCGGGTC		
*IL-8*	F-GCAAGGTAGGACGCTGGTAA	107	NM_205498.2
	R-GCGTCAGCTTCACATCTTGA		
*TGF-β4*	F-CGGGACGGATGAGAAGAAC	258	M31160.1
	R-CGGCCCACGTAGTAAATGAT		
*ZO-1*	F-CTTCAGGTGTTTCTCTTCCTCCTC	131	XM_413773.4
	R-CTGTGGTTTCATGGCTGGATC		
*Claudin-1*	F-CATACTCCTGGGTCTGGTTGGT	100	AY750897.1
	R-GACAGCCATCCGCATCTTCT		
*Claudin-2*	F-CAACTGGAAGATCAGCTCCT	119	NM_001277622.1
	R-TGTAGATGTCGCACTGAGTG		
*Occludin*	F-ACGGCAGCACCTACCTCAA	123	D21837.1
	R-GGGCGAAGAAGCAGATGAG		
*Mucin-2*	F-TTCATGATGCCTGCTCTTGTG	93	XM_421035.2
	R-CCTGAGCCTTGGTACATTCTTGT		

F Upstream primer, R Downstream primers. ^a^ The primers were synthesized by Shanghai Shenggong Biotechnology Co., Ltd. (Shanghai, China)

**Table 3 animals-14-00955-t003:** Effects of TA on jejunal antioxidant capacity of broilers co-infected with CCP.

Item	CON	CCP	CTA	*p*-Value
Day 14				
GSH-Px(U/mg prot)	21.97 ± 4.40 ^b^	16.90 ± 4.45 ^c^	26.96 ± 2.03 ^a^	<0.001
T-SOD (U/mg prot)	192.04 ± 19.56	181.88 ± 13.81	180.72 ± 28.72	0.322
CAT (U/mg prot)	1.31 ± 0.12 ^a^	0.82 ± 0.23 ^c^	1.13 ± 0.10 ^b^	<0.001
H_2_O_2_ (mmol/g prot)	1.88 ± 0.22 ^b^	3.03 ± 0.29 ^a^	2.01 ± 0.40 ^b^	<0.001
MDA (nmol/mg prot)	7.15 ± 1.42 ^b^	10.52 ± 1.11 ^a^	9.37 ± 3.06 ^a^	<0.001
T-AOC (mmol/g prot)	0.11 ± 0.01	0.10 ± 0.01	0.10 ± 0.01	0.052
Day 21				
GSH-Px (U/mg prot)	28.82 ± 3.75 ^b^	24.12 ± 2.17 ^c^	31.18 ± 2.76 ^a^	<0.001
T-SOD (U/mg prot)	178.16 ± 12.25 ^ab^	167.95 ± 11.94 ^b^	184.79 ± 18.14 ^a^	0.013
CAT (U/mg prot)	2.27 ± 0.39	2.30 ± 0.32	2.37 ± 0.35	0.747
H_2_O_2_ (mmol/g prot)	6.16 ± 0.50 ^b^	7.62 ± 1.94 ^a^	5.38 ± 1.35 ^b^	0.001
MDA (nmol/mg prot)	6.09 ± 0.91 ^b^	7.03 ± 1.39 ^a^	5.83 ± 1.14 ^b^	0.024
T-AOC (mmol/g prot)	0.15 ± 0.01	0.15 ± 0.02	0.16 ± 0.02	0.084
Day 28				
GSH-Px (U/mg prot)	68.42 ± 9.34	63.18 ± 9.80	64.33 ± 13.58	0.428
T-SOD (U/mg prot)	364.21 ± 21.01 ^a^	313.83 ± 32.65 ^c^	339.47 ± 28.04 ^b^	<0.001
CAT (U/mg prot)	6.32 ± 0.99 ^a^	5.07 ± 0.38 ^b^	5.85 ± 0.81 ^a^	<0.001
H_2_O_2_ (mmol/g prot)	2.58 ± 0.60	3.08 ± 0.76	2.64 ± 0.50	0.086
MDA (nmol/mg prot)	5.85 ± 1.63	7.58 ± 1.56	7.12 ± 1.56	0.076
T-AOC (mmol/g prot)	0.17 ± 0.05 ^a^	0.12 ± 0.01 ^b^	0.17 ± 0.06 ^a^	0.028

Numbers are expressed as mean and SD, n = 14. ^a,b,c^ Values not sharing a superscript in the same row are markedly different.

## Data Availability

The datasets used and/or analyzed during the current study are available from the corresponding author upon reasonable request.

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
