# Peer review of "Effect of Tannic Acid on Antioxidant Function, Immunity, and Intestinal Barrier of Broilers Co-Infected with Coccidia and Clostridium perfringens"

_animals, 2024, doi:10.3390/ani14060955_

Round 1

Reviewer 1 Report

Comments and Suggestions for Authors

1.        Spell out “AA” as arbor acres at the fist appearance.

2.        Reanalyze the data using another statistical method, because Duncan’s test is not an appropriate for this scientific area.

3.        It is necessary to set non-infection and TA-supplemented group for precisely evaluating the TA effect. It is difficult to evaluate whether the alleviating effect of TA was observed only for C/CP co-infection induced disorders.

4.        Blood biochemical analyses was normalized to protein values. Immuno-stimulation may change blood protein concentration. How about the blood protein content?

5.        Discuss how TA affect the plasma parameters and hepatic inflammation in terms of the bioavailability of TA.

Author Response

Thank you for your suggestion. Please refer to the attachment for my response

Reviewer 2 Report

Comments and Suggestions for Authors

I am pleased to participate in reviewing this interesting research manuscript. The study investigated the mitigatory effect of tannic acid supplementation in diet on some necrotic enteritis deleterious pathological outcomes in the liver and jejunum of broiler chickens. However, the necrotic enteritis model was established using a well-documented approach, the necrotic enteritis model has not been tested to confirm its success before proceeding with the research. The method section needs major improvement (details are provided). Most of the results are consistent with previous studies using Tannic acid with the same necrotic enteritis model in broiler chickens. However, the posted conclusion that TA improved antioxidant capacity through the Nrf2-Keap1 pathway to protect the intestinal health of broilers challenged with CCP, should be revised because the results do not support the part of the conclusion (… through the Nrf2-Keap1 pathway….).

Line (L) 16 and 54: “Necrotizing enterocolitis (NEC)” is a different condition (Ginglen, J. G., & Butki, N. (2018). Necrotizing enterocolitis.; https://www.ncbi.nlm.nih.gov/books/NBK513357/). Please change to Necrotic enteritis (NE).

L 21: “…. function in NE infected broilers..” NE is a disease condition, not an infection. Please change into a correct expression such as “…. function in broilers with NE condition..”.

L 48: “….attenuated the inflammatory response..”, I think “attenuated…..response” is not a suitable word because it sounds like affects the inflammation in a way that weakens the immune response. I believe using “mitigated the jejunal inflammation …” will be a better expression

L 55-56 and 226-229: “…together or secondary infected by coccidia,….” Incorrect information, please rewrite. Chicken necrotic enteritis is developed upon predisposing (primary) factors including Eimeria spp infection (mainly Eimeria maxima), intestinal mucosa injury, or dietary (such as High dietary amounts of animal by-products, which promote the growth and colonization of Clostridium perfringens. Also, co-infection is possible, as you mention (together). So, Clostridium perfringens is secondary to coccidia (coccidia is primary), or they may co-infected.

L 65: “….exist….”, Do you mean exhibit?

L 68-69: “…avoid compensatory liver hypertrophy in broilers..”, Please provide a specific reference for the positive effect of tannins on preventing compensatory liver hypertrophy in broilers. Otherwise, remove this part of the sentence.

L 71: What are the “intestinal compact proteins”, Do you mean “intact” instead of “compact”.

L 71: You used references for studies that were performed in mammalians (i.e., mice) (references 11-13). Regarding the physiological and anatomical differences, please provide references for research that was conducted in avian. You may keep the current references with the avian references, but, you should mention the animal used in this research, if it is not chicken/avian, such as when you mentioned “in rats, line 73”.

L 71-72: Again “attenuate” sounds unsuitable, you may use “mitigate, alleviate, or assuage.”

L 73-75: “However, the impact of tannins on gut health and anti-oxidative functions in NE infected broilers is not completely clarified.” This is not true. Since 2016, there have been five published research articles investigating the effect of tannins on necrotic enteritis/ Eimeria and C. perfringens co-infection.

L 30: Identify the abbreviation (AA) as “Arbor Acres” at first use.

L 92-93: Mention the vaccine composition (Eimeria spp composition, oocyst count for each spp) and the commercial name. It is not clear how much was the administrated dose compared to the recommended dose? Have you used 30 times the recommended dose? Please review and rewrite more clearly.

L 95: Did you use gavage for oral administration of C. perfringens, or any oral other oral administration? Please explain.

L 103-107: Which part of the jejunum did you sample?

L 115- 116: Something is missing here. Did you homogenize the tissues? What was pelleted and why, if the tissue was intact? Notice “S” is capitalized after a comma in the sentence “….is pipetted, Samples were…”?!

L 123: “….formed by Guo and Li…”, Change “by” to “as reported in”

L 125: Reference (20) is not the original reference for the 2-ΔΔCt method.

            You may use: Fu, W. J., Hu, J., Spencer, T., Carroll, R., & Wu, G. (2006). Statistical models in assessing fold change of gene expression in real-time RT-PCR experiments. Computational biology and chemistry30(1), 21-26.”

If it is correct, please mention the model because this article tested three models.

            However, whether you used the above reference or not, you should cite the original reference for the 2-ΔΔCt method: “Livak, K. J., & Schmittgen, T. D. (2001). Analysis of relative gene expression data using real-time quantitative PCR and the 2− ΔΔCT method. methods25(4), 402-408.”

L 121-127: Did you use SYBR Green dye as a probe for the qPCR? Or different dye was used? Please explain?

Table 2: Please provide the product length.

How did you confirm that the CCP and CTA were successfully infected?

Results: The mortality and clinical status data should be reported.

L 265-267: “….suggested that TA could enhance antioxidant enzyme activities through the Nrf2-Keap1 pathway to alleviate oxidative damage in the liver and jejunum caused in the CCP challenged broilers.” There was not enough evidence to address this suggestion. In the liver, the CTA group showed only Nrf2 upregulation at d 14 without upregulation of Keap1, and without upregulation of the downstream antioxidant defense factors-associated genes HO-1 and NQO1, while at 21 and 28 days Keap1 and Nrf2 genes showed lower expression levels in CTA group than CCP. The results of the jejunum tissue were similar, the CTA group showed lower expression levels of Keap1 and Nrf2 genes and either no difference or lower expression levels of the downstream antioxidant defense factors-associated genes compared with the CCP group. Furthermore, the expression levels of Keap1 and Nrf2 genes do not solely govern the Nrf2-Keap1 pathway activation, but also the ubiquitination and degradation of transcription factor Nrf2 equally controls its function. Therefore, further investigations are required to provide plausible evidence supporting the suggestion that “TA could enhance antioxidant enzyme activities through the Nrf2-Keap1 pathway”. Such as measuring the proteins (and ubiquitinated Nrf2) levels using western blot, and co-precipitating Nrf2-Keap1 protein complex, etc.

L 270-271: Please add “certain immune-related genes.”

L 274-275: Upon intestinal infection, a significant elevation in the expression of the pro-inflammatory genes, may or may not coincide with a reduction in the anti-inflammatory genes’ expression. Please rewrite accordingly.

Figure captions: Identify the panels “A, B, and C” in the caption (e.g., (A) on day 14, etc..)

Mention the statistical analysis method (i.e., One-way 133 ANOVA/ Duncan’s multiple-range test)

Conclusion: Again change “attenuate” the inflammatory with any previously suggested words.

Conclusion: There was not enough evidence to conclude “….through the Nrf2-Keap1 pathway to protect the intestinal health of broilers challenged with CCP.” Without further experiments/investigation, you have to rewrite with an expression implying a huge degree of uncertainty (for example, the results may indicate that TA could improve antioxidant capacity through different mechanisms including activating the transcriptional factor Nrf2 downstream of the Nrf2-Keap1 pathway.

Author Response

(The authors gave the same response as above.)

Reviewer 3 Report

Comments and Suggestions for Authors

This paper examines the effects of tannic acid on intestinal conditions using CPP model. In this study, Clostridium perfringens (CP) and coccidia were used as models of necrotizing enterocolitis. I have some doubts about this article, so I would like to have them clarified. I will raise the points below:

Point 1: The volume of TA in this study was 1 g/kg. Is this amount relative to 1 kg of body weight or to the amount of feed? Is this an appropriate feeding amount?  Regarding the effects of TA, several genes examined in the liver appear to be exacerbated, especially GST, Nrf2, and GSH-Px3 in d28. There are concerns about the negative effects of TA on the liver. Then, there should also be a description of the body weight changes of the broilers during the experimental period.

Point 2: In this study, is there any evaluation of intestinal tissue damage due to CCP inoculation? Since Cp is a typical bacterium that causes necrotizing enterocolitis, it is thought that there is tissue damage. However, I think it is necessary to proceed with the discussion after verifying the tissue damage caused by CP and observing the symptoms. CP is certainly the typical pathogen of necrotizing enterocolitis, but is the jejunum an appropriate site for tissue analysis? Liver and jejunum are used as tissues for gene expression analysis, but the entire intestine should not be considered either. Is not it? Additionally, in this experiment, not only CP but also coccidia vaccine strains were used. The liver and jejunum are often used as tissues for gene expression analysis, but shouldn't the whole intestine also be considered?

Point 3: This paper uses coccidia and clostridium as models of NE. It is stated that a quadrivalent vaccine was used for coccidia. I think that the effects on tissues may be different between highly virulent and attenuated strains, but why not use highly virulent strains? We also think that the quadrivalent vaccine used in this study should clearly state what protozoan species it contains. It is easy to understand if the research is aimed at the immune-enhancing effect of tannic acid. If the research was aimed at the immune-enhancing effect of tannic acid, I think it would be easy to understand why a vaccine strain was used.

Mainor:  Since not all researchers are familiar, it is better to specify the difference GSX-Px1 and GSX-Px3.

Author Response

(The authors gave the same response as above.)

Round 2

Reviewer 1 Report

Comments and Suggestions for Authors

1.         Regarding Comment #3, the content should include the Discussion.

2.         Regarding Comment #4, the data that has already been published should also include the discussion as a sentence.

3.         Regarding Comment #5, the reply is not enough. How much of the TA ingested was absorbed and circulated in the body? These “bioavailability” of TA will be low, the main active site could be intestine. For this point, the authors need to discuss.  

Author Response

Thank you for your suggestion, please refer to the attachment for details

Reviewer 2 Report

Comments and Suggestions for Authors

Please mention the post hoc test used for multiple comparisons. You should have used one?!

Author Response

(The authors gave the same response as above.)

Reviewer 3 Report

Comments and Suggestions for Authors

I believe appropriate modifications have been made based on review comments.

Author Response

Dear honored reviewer,

Thank you for recognizing us, we will work harder in the future! Again, thank you for your kindness and patience.

Best regards!

Yours Sincerely,

Z.F. Zhang